# BALANCING SYMMETRY AND EFFICIENCY IN GRAPH FLOW MATCHING

**Benjamin Honoré, Alba Carballo-Castro, Yiming Qin, Pascal Frossard**[*]
LTS4, EPFL, Lausanne, Switzerland

## ABSTRACT

Equivariance is central to graph generative models, as it ensures the model respects the permutation symmetry of graphs. However, strict equivariance can increase computational cost due to added architectural constraints, and can slow down convergence because the model must be consistent across a large space of possible node permutations. We study this trade-off for graph generative models. Specifically, we start from an equivariant discrete flow-matching model, and relax its equivariance during training via a controllable symmetry modulation scheme based on sinusoidal positional encodings and node permutations. Experiments first show that symmetry-breaking can accelerate early training by providing an easier learning signal, but at the expense of encouraging shortcut solutions that can cause overfitting, where the model repeatedly generates graphs that are duplicates of the training set. On the contrary, properly modulating the symmetry signal can delay overfitting while accelerating convergence, allowing the model to reach stronger performance with 19% of the training epochs.

## 1 INTRODUCTION

The emergence of diffusion models has redefined generative modeling, showing strong performance in capturing complex data distributions through iterative denoising (Sohl-Dickstein et al., 2015; Song & Ermon, 2019; Ho et al., 2020). This success has also been observed in graph generative models, where diffusion- and flow matching-based approaches achieve state-of-the-art results. Recent work suggests that this is related to implicit dynamical regularization during training, which discourages memorization and instead promotes learning the underlying structural patterns of the data (Bonnaire et al., 2025). In graph-structured data, this behavior is closely related to permutation equivariance. Since reordering node indices does not change the graph itself, a model must produce the same output for different node orderings. Enforcing this property through equivariant architectures has been shown to yield a strict generalization benefit (Elesedy & Zaidi, 2021).

However, restricting the hypothesis space to strictly equivariant functions can be limiting. Equivariant models can be criticized in terms of computational overhead and scalability due to their added architectural constraints (Nowara et al., 2025). Moreover, the equivariance hypothesis on geometrical data introduces additional structure into the learning process, which increases optimization complexity and can slow convergence speed during training (Manolache et al., 2025). Beyond equivariant model architectures, recent work shows that domain symmetries can also be learned dynamically through data augmentation, reducing the need to enforce them in the architecture (Brehmer et al., 2025). These observations suggest a trade-off between symmetry awareness and computational efficiency: while strict equivariance provides theoretical guarantees and generalization benefits, overly tight constraints can hinder practical training efficiency. This motivates intermediate solutions, such as restoring invariance only at inference time (Yan et al., 2024), or relaxing equivariance using sinusoidal positional encodings (PE) that assign each node a unique index, allowing the model to distinguish between nodes but avoiding strict equivariance constraints (Lawrence et al., 2025).

We investigate this phenomenon by studying the role of PE in the learning dynamics of graph generative models. Specifically, we consider DeFoG (Qin et al., 2025), a discrete flow model on graphs with an equivariant backbone. We contrast sinusoidal positional encodings that explicitly break symmetry with structure-aware alternatives which preserve equivariance by construction such as RRWP

---

[*]Contact: `first_name.last_name@epfl.ch`

(Ma et al., 2023). Moreover, we analyze different strategies for modulating symmetry-breaking signals, including positional encodings with different scales and in-training node permutations. Experiments show that symmetry breaking accelerates early training by providing an easier learning signal, but also promotes shortcut solutions that lead to overfitting, such as repeatedly generating training graphs. In contrast, properly modulated symmetry breaking reduces optimization difficulty and accelerates convergence while delaying overfitting in early training.

## 2 ASSESSING THE RELEVANCE OF POSITIONAL ENCODINGS

Our study is motivated by the hypothesis that relaxing strict equivariance through symmetry breaking positional encodings (PEs) can simplify the learning space and thus improve training efficiency, at the cost of generalization. Within the discrete flow matching framework for graph generation (see Appendix B for theoretical details of this framework), we investigate this trade-off between symmetry preservation and training efficiency in detail. Our analysis focuses on unlabeled graphs.

### 2.1 MECHANISMS FOR SYMMETRY MODULATION

First, we design a tunable symmetry modulation mechanism to adjust the degree of symmetry breaking, thereby enabling a systematic analysis of its effect on optimization speed and generalization behavior. We consider two approaches: scaled positional encodings and in-training permutations. A third approach using data augmentation is discussed in Appendix E.

**Symmetry breaking via scaled positional encodings.** We control symmetry breaking by modifying the positional encodings assigned to each node. Let $p \in \mathbb{R}^{n,d}$ denote the positional encoding, and $p_i$ denotes the encoding of node $i$, constructed using fixed sinusoidal functions with $d$ channels. This encoding is independent of the graph structure (see Appendix C for further details). We define the modulated positional encoding as

$$p_i(\lambda) = \lambda \langle p \rangle_i + (p_i - \langle p \rangle_i), \tag{1}$$

where $\langle p \rangle_i = \langle p_i \rangle \cdot (1, \ldots, 1)^T$ denotes the permutation-invariant component obtained by averaging the positional encodings across nodes, and $\lambda \in \mathbb{R}^+$ is a scalar control parameter. This decomposition separates invariant and non-invariant contributions and allows $\lambda$ to continuously modulate the strength of the symmetry-breaking signal introduced by the positional encodings, attenuating the signal as $\lambda$ increases.

**Symmetry restoration via in-training permutations.** To counteract symmetry breaking and encourage symmetry awareness during training, we additionally apply random permutations to the input graphs. We define $\chi$ as the number of training epochs between successive permutations. Smaller values of $\chi$ enforce more frequent permutations and stronger symmetry restoration, while larger values allow the model to exploit asymmetry over longer training intervals. This mechanism provides a complementary control lever that reintroduces symmetry without modifying the model architecture.

### 2.2 DYNAMICS OPTIMIZATION

We now combine the mechanisms previously introduced to explore the trade-off between symmetry awareness and optimization efficiency, as illustrated in Figure 1. In particular, we will explore the Stochastic Block Model (SBM) dataset (Martinkus et al., 2022) and test different settings in the $(\lambda, \chi)$ space.

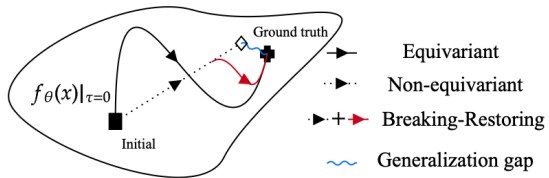

Figure 1: **Symmetry breaking restoring cycle.** Starting from the initial point, equivariant paths converge slowly but preserve symmetry, non equivariant paths converge faster with a generalization gap, and the breaking restoring path achieves fast progress while recovering structural validity.

Specifically, we study the effect of different positional encodings $p(\lambda)$, $\lambda \in \{1, ..., 5\}$ without permutations ($\chi = \infty$). A similar experiment is then carried out by setting the permutation rate $\chi = 10$ and testing again different values of $\lambda$. Finally, we experiment with a proposed symmetry breaking restoring cycle (Figure 1), where PEs are used to escape the initial convergence bottleneck, and then permutation symmetry is dynamically restored to maintain sampling quality (Figure 3).

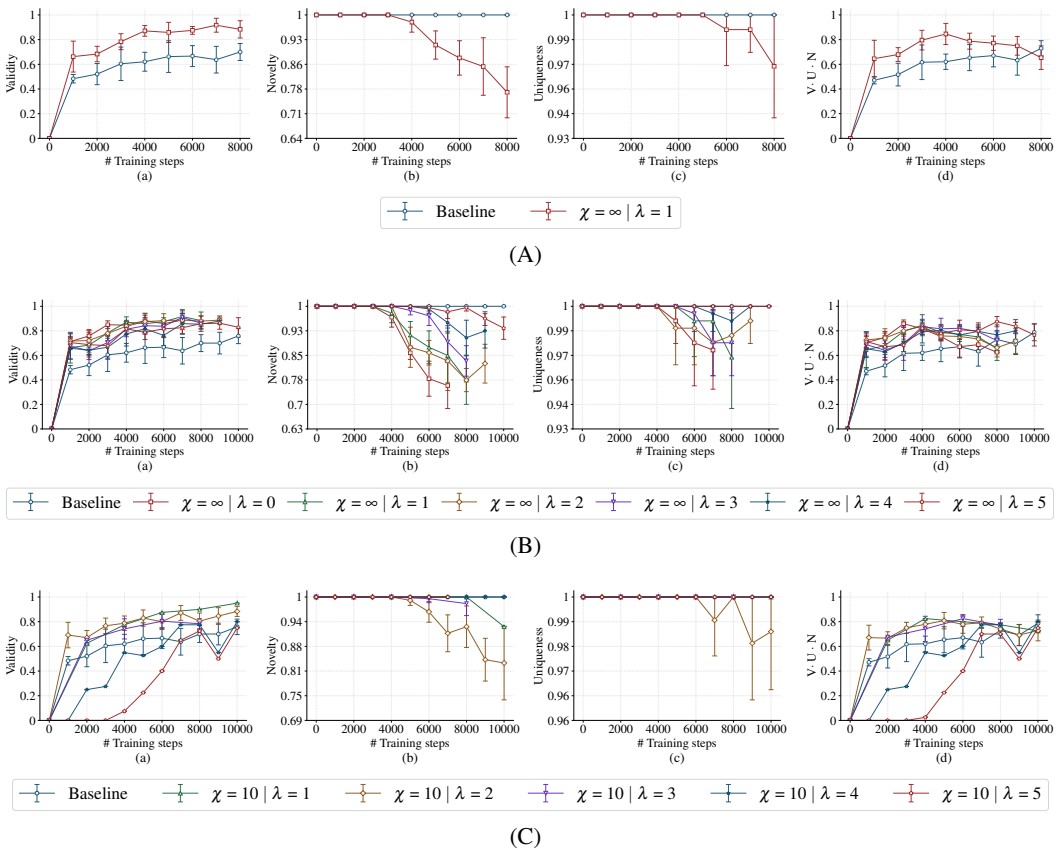

Figure 2: Comparison of the baseline setting and symmetry breaking sinusoidal positional encodings in (A), and with different invariance scalings $\lambda$ using (B) $\chi = \infty$ and (C) $\chi = 10$. Panels show (a) Validity, (b) Novelty, (c) Uniqueness, and (d) VUN on the *SBM* dataset.

**Evaluation metrics.** We evaluate our study by measuring the impact of the different modifications on the validity, uniqueness, and novelty of the generated graphs through the VUN metric. Validity (V) is the proportion of generated graphs that satisfy domain constraints; Uniqueness (U) measures the fraction of valid graphs that are not isomorphic to other generated samples; and Novelty (N) is the fraction of valid graphs not present in the training set. We interpret U and N as indicators of memorization, and V as a proxy for convergence (see Appendix C.2). Overall, a high VUN score indicates that the model learns the target graph structure and generalizes beyond the training data.

## 3 EXPERIMENTAL RESULTS

We analyze the trade-off introduced by symmetry breaking positional encodings in graph generation. We show first sinusoidal encodings accelerate early convergence but also lead to earlier overfitting, reflected by reduced novelty and uniqueness. We then study how to modulate this trade-off with positional encoding scaling, and how to restore generalization with permutations. Finally, we compare the best performing configurations to the baseline DeFoG model with structure-aware encodings, namely random walk features such as RRWP.

**Effect of different positional encodings.** We first observe in Figure 2(A) that, compared to equivariant RRWP encodings (baseline), symmetry breaking sinusoidal positional encodings improve early validity, indicating faster convergence, but reduce novelty and uniqueness, revealing a trade-off between training speed and generalization on *SBM* (see Appendix G for results on other datasets).

These observations motivate further modulation of the symmetry breaking signal during training to delay the onset of overfitting while retaining faster convergence. In Figure 2(B), we observe that larger values of $\lambda$, which emphasize the order-agnostic signal and strengthen symmetry preservation,

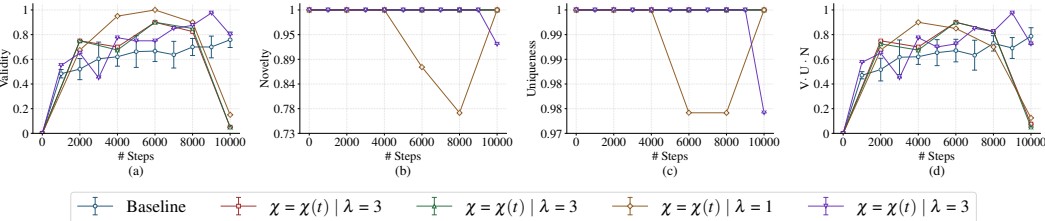

Figure 3: Comparison of DeFoG's baseline setting, with RRWP encodings and symmetry breaking sinusoidal PE with different invariance scaling $\lambda$ and time dependent permutation rates.

delay the collapse of uniqueness and novelty, whereas smaller values accelerate early convergence at the cost of earlier overfitting. The validity curves remain consistent with the optimization dynamics discussed in Section 2.2. This confirms that $\lambda$ is an effective control parameter for regulating the trade-off between convergence speed and generalization.

We next examine whether introducing explicit permutations can further delay overfitting beyond what is achieved by positional encoding scaling alone. By introducing permutations with a fixed rate $\chi = 10$, results in Figure 2(C) show that the collapse of uniqueness and novelty is further delayed. However, when combined with larger values of $\lambda$, permutations also tend to slow convergence, highlighting a tension between increased symmetry preservation and training efficiency.

**Symmetry breaking-restoring cycle.** We further consider whether increasing the permutation frequency over training can progressively drive the system toward equivariance and further mitigate overfitting. Using time dependent permutation rate functions, which apply graph permutations at varying frequencies during training, we observe the existence of a permutation threshold above which uniqueness and novelty, if degraded during early training, can be recovered (Figure 3). However, enforcing a sudden transition back to full equivariance leads to a sharp degradation in performance. In contrast, smoothly increasing the permutation rate mitigates this effect, allowing generalization to be restored and providing the best performances obtained with this method. Further details on the choice of rate functions are provided in Appendix D.

**Best performing configurations.** The best performances obtained with the configurations described in Section 2 are shown in Table 1. For each configuration, we report the maximal value of the VUN metric and a metric of structural validity (Average ratio), as well as the training epoch where this value is attained. The best performance is obtained with a time dependent permuta-

Table 1: Optimal configurations obtained using the presented methods, ordered by VUN performances.

| $\lambda$ | $\chi$ | VUN ↑ | Avg Ratio ↓ | Epoch | Epochs / Min ↑ |
|---|---|---|---|---|---|
| 3 | $\chi(t)$ | 0.975 | 1.25 | 9000 | 11.96 |
| 1 | $\infty$ | 0.925 | 2.01 | 4000 | 11.93 |
| 3 | 5 | 0.925 | 3.16 | 10000 | 11.71 |
| 5 | $\infty$ | 0.900 | 2.36 | 6000 | 12.01 |
| Baseline | $\infty$ | 0.900 | 2.24 | 21000 | 10.47 |
| 1 | 10 | 0.850 | 1.63 | 12000 | 11.89 |

tion rate, which indicates that the proposed framework achieves better performance than the baseline with less training epochs.

Owing to the simplicity and graph independence of sinusoidal positional encodings, we also observe a reduced computational cost during training. The last column of Table 1 reports the average number of training epochs per minute, including sampling time. Within our setting, we can reach better performance, than the baseline with $19\%$ of the training steps.

## 4 CONCLUSION

We study equivariance breaking in graph generative models as a controlled design choice rather than a universal improvement. Our experiments demonstrate clear benefits on the *SBM* dataset, where sinusoidal positional encodings outperform graph-based RRWP encodings while requiring fewer training steps, provided that equivariance is relaxed in a controlled manner. These improvements are less pronounced on simpler datasets such as planar graphs and trees, where our results indicate that in such regimes equivariance remains a key factor for maintaining generalization. Overall, these findings position controlled equivariance breaking as a complementary mechanism (most effective in complex settings) rather than a replacement for equivariant inductive biases.

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

## A  RELATED WORK

**Graph generative models**  A lot of recent work on graph generative modeling has increasingly focused on diffusion-based and flow-based approaches, which generate graphs through iterative denoising or continuous transport processes. Early graph diffusion models adapted continuous state-space diffusion frameworks to graph-structured data, including EDP-GNN (Niu et al., 2020), DGSM (Luo et al., 2021), and GeoDiff (Xu et al., 2022). Subsequent methods extended diffusion to discrete state spaces, with DiGress (Vignac et al., 2023) and MCD (Haefeli et al., 2022) operating in discrete time. Continuous-time formulations were later introduced, encompassing both continuous and discrete state spaces, such as GDSS and GruM (Jo et al., 2022; 2024), as well as DisCo and Cometh (Xu et al., 2024; Siraudin et al., 2024). More recently, DeFoG (Qin et al., 2025) has achieved state-of-the-art performance by leveraging discrete flow matching, connecting diffusion-based generative modeling with optimal transport–inspired formulations.

**Equivariant networks and symmetry breaking**  Permutation equivariance is a central inductive bias in graph neural networks, ensuring that model outputs are invariant to node reindexing and leading to consistent predictions and provable generalization gains Elesedy & Zaidi (2021). However, exact equivariance also imposes a fundamental limitation: equivariant functions are unable to break symmetries at the level of individual data samples, which can restrict expressivity and hinder optimization (Kaba & Ravanbakhsh, 2024). These constraints are further reflected in practical concerns, including increased computational overhead and reduced flexibility in tasks where partial or data-driven symmetry breaking is desirable (Lawrence et al., 2025; Nowara et al., 2025). Complementary evidence from the physical sciences suggests that unconstrained models can often learn symmetries implicitly from data, with small symmetry violations having negligible impact on predictive accuracy and sometimes improving convergence behavior (Langer et al., 2024). To mitigate these limitations, recent work has explored softer alternatives to architectural equivariance, including learning symmetries through data augmentation (Brehmer et al., 2025) and introducing controlled symmetry breaking via positional encodings or inference-time corrections (Yan et al., 2024; Lawrence et al., 2025).

## B  DEFOG: DISCRETE FLOW MATCHING FOR GRAPH GENERATION

DeFoG (Qin et al., 2025) leverages *Discrete Flow Matching* (DFM) (Gat et al., 2024; Campbell et al., 2024) to generate graphs by operating directly in the categorical state space. It defines a stochastic flow that gradually transforms samples from a simple noise distribution into structured graphs, enabling expressive modeling of global graph dependencies while retaining tractable training and flexible sampling dynamics.

### B.1  MATHEMATICAL FRAMEWORK

We define a directed graph as $G = (\mathbf{x}, \mathbf{e})$, where $\mathbf{x} = \{x^{(n)}\}_{n=1}^N$ represents $N$ categorical nodes and $\mathbf{e} = \{e^{(i,j)}\}_{1 \le i \ne j \le N}$ represents the set of edges. Nodes can take categorical values in $\{1, \ldots, X\}$ and edges in $\{1, \ldots, E\}$, where one edge class typically denotes the absence of a connection. In this work, we focus on unlabeled graphs, and therefore only consider a single node class and two edge classes (presence or absence of the edge). Finally, the subscript $t \in [0, 1]$ denotes the state of the graph during the generative flow.

### B.2  FORWARD (NOISING) PROCESS

The forward process gradually perturbs a clean graph $G_1$ into noise as time decreases from $t = 1$ to $t = 0$. Corruption is applied independently to each discrete variable via linear interpolation between the data distribution and the noise distribution (Gat et al., 2024; Campbell et al., 2024).

For a node variable $\mathbf{x}^{(n)}$, the conditional distribution at time $t$ is

$$p_{t|1}^X\left(\mathbf{x}_t^{(n)} \mid \mathbf{x}_1^{(n)}\right) = t\,\delta\left(\mathbf{x}_t^{(n)}, \mathbf{x}_1^{(n)}\right) + (1 - t)\,p_{\text{noise}}^X\left(\mathbf{x}_t^{(n)}\right), \qquad (2)$$

where $\delta(\cdot, \cdot)$ denotes the Kronecker delta and $p_{\text{noise}}^X$ is a fixed categorical prior over node labels. An analogous construction is used for each edge variable $\mathbf{e}^{(i,j)}$ (Qin et al., 2025). The resulting

marginal distribution at each timestep $\boldsymbol{p}_t(G_t \mid G_1)$ defines a tractable family of intermediate graph distributions indexed by $t$.

## B.3   REVERSE (DENOISING) PROCESS

Generation proceeds by reversing the noising dynamics, starting from a sample $G_0 \sim \boldsymbol{p}_0$ and evolving toward $t = 1$. This reverse process is formulated as a continuous-time Markov chain (CTMC) with transition kernel

$$\boldsymbol{p}_{t+\Delta t \mid t}(G_{t+\Delta t} \mid G_t) = \boldsymbol{\delta}(G_t, G_{t+\Delta t}) + \boldsymbol{R}_t(G_t, G_{t+\Delta t}) \, \mathrm{d}t, \tag{3}$$

where $\boldsymbol{R}_t$ is the rate matrix governing discrete state transitions (Campbell et al., 2024).

In practice, the CTMC is simulated using a time discretization with step size $\Delta t$ and an Euler-style update. Since the true rate matrix is unknown, it is approximated by a learned estimator $\boldsymbol{R}_t^\theta$, derived from a neural network that predicts the posterior distribution of the clean graph given $G_t$.

The denoising model outputs factorized predictions over nodes and edges,

$$\boldsymbol{p}_{1\mid t}^\theta(\cdot \mid G_t) = \left( \{\boldsymbol{p}_{1\mid t}^{\theta,(n)}(\mathbf{x}_1^{(n)} \mid G_t)\}_{n=1}^N, \{\boldsymbol{p}_{1\mid t}^{\theta,(i,j)}(\mathbf{e}_1^{(i,j)} \mid G_t)\}_{i \neq j} \right), \tag{4}$$

which are then used to construct $\boldsymbol{R}_t^\theta$.

Training minimizes a cross-entropy objective applied independently to all node and edge variables:

$$\mathcal{L} = \mathbb{E}_{t,G_1,G_t}\left[ -\sum_n \log \boldsymbol{p}_{1\mid t}^{\theta,(n)}\left(\mathbf{x}_1^{(n)} \mid G_t\right) - \lambda \sum_{i \neq j} \log \boldsymbol{p}_{1\mid t}^{\theta,(i,j)}\left(\mathbf{e}_1^{(i,j)} \mid G_t\right) \right], \tag{5}$$

where $t$ is sampled from a predefined distribution on $[0,1]$, $G_1 \sim \boldsymbol{p}_1$ is a clean training graph, and $G_t \sim \boldsymbol{p}_t(\cdot \mid G_1)$ is its corrupted counterpart. The scalar $\lambda > 0$ balances node- and edge-level supervision.

## B.4   SAMPLING

A key advantage of discrete flow matching is that the learned denoising model can be combined with modified sampling dynamics at inference time. Qin et al. (2025) propose three mechanisms to control the behavior of the generative trajectory at sampling time:

**Target-Guidance.**   Sampling can further be biased toward the predicted clean graph by augmenting the conditional rate matrix with an explicit guidance term. Concretely, the guided rate is defined as

$$R_t(z_t, z_{t+\Delta t} \mid z_1) = R_t^*(z_t, z_{t+\Delta t} \mid z_1) + \omega \cdot \frac{\delta(z_{t+\Delta t}, z_1)}{\mathcal{Z}_t^{>0} \, p_{t\mid 1}(z_t \mid z_1)}, \tag{6}$$

where $\omega \geq 0$ controls the guidance strength and the additional term increases the probability of transitions that directly match the clean target configuration.

**Stochasticity.**   Finally, the level of randomness in the denoising trajectory can be tuned by adding a detailed-balance-preserving auxiliary rate matrix $R_t^{\mathrm{DB}}$:

$$R_t^\eta = R_t^* + \eta \, R_t^{\mathrm{DB}}. \tag{7}$$

Larger values of $\eta$ encourage exploration and can help escape suboptimal intermediate states, while $\eta = 0$ recovers the minimum-variance, more deterministic flow.

**Time Distortion.**   Sampling can be altered by applying a monotone bijection $f : [0,1] \to [0,1]$ to the time variable, yielding a distorted schedule $t' = f(t)$. While time distortions can also be used during training to bias learning toward specific noise levels, at sampling time they primarily control the effective step size along the denoising path.

If $t$ is uniformly distributed, the induced density of $t'$ is

$$\phi_{t'}(t') = \left| \frac{d}{dt'} f^{-1}(t') \right|. \tag{8}$$

We consider the same family of distortion functions as Qin et al. (2025), including identity, polynomial, and cosine-based mappings, which emphasize different regions of the trajectory.

## C EXPERIMENTAL DETAILS

We first detail the positional encodings used in Section 2, and then describe the datasets used in our experiments and outline the according performance metrics. All experiments were run on a NVIDIA A100-SXM4-80GB GPU.

### C.1 POSITIONAL ENCODINGS

We first describe the two main types of positional encodings used in this study:

**RRWP encodings** In our baseline setting, we use DeFoG together with graphs nodes augmented by *Relative Random Walk Probabilities* (RRWP). These encodings are based on random walk probabilities, and computed in the following way (Ma et al., 2023). Let us consider a graph $\mathcal{G}$ and its adjacency matrix $A(G)$, and its *degree matrix*, defined as $D_{i,j} = \delta_{i,j} \deg(v_i)$, where deg counts the number of edges terminated at the node $v_i$. We then compute $M = D^{-1}A$, whose elements $M_{i,j}$ are the transition probabilities of node $i$ jumping to node $j$ in a simple random walk process. Indeed, the adjacency matrix contains the information on all nodes connections, and $D^{-1}$ acts as a mere normalization factor. We can then easily compute the $K$-step transition from node $i$ to $j$, by taking powers of $M$, $M_{i,j}^K$. We finally define the RRWP encodings as

$$P_{i,j}^K = [\delta_{i,j}, M_{i,j}, M_{i,j}^2, ..., M_{i,j}^K]. \tag{9}$$

This construction is analogue to Ma et al. (2023), Qin et al. (2025).

A crucial result from Ma et al. (2023), is that they show that using RRWP together with an MLP architecture is expressive. In particular, using $K$-hop RRWP encodings reveals higher-order structures in a graph.

**Sinusoidal positional encodings** In contrast with the structure-aware encodings presented above, we introduce "absolute" positional encodings. In this regard, these encodings are not defined based on the structural properties of the graph, such as its adjacency matrix, as it was the case for RRWP. Instead, they define an absolute frame on a graph, regardless of the edge structure, and different scales contained in the topology of the graph. Equipping graph's nodes with such encodings is then inconsistent with the symmetric structure of a graph. In particular, there should not be a well defined notion of node ordering on a graph, which is precisely the information contained in such encodings. However, they are still appealing as they require less compute, which together with equivariance breaking, motivates their use in this study.

We introduce *sinusoidal positional encodings*, which take the vector $p_i = i$ and encode it over several channels, using the trigonometric functions sine and cosine. Accordingly, we define

$$p_{i,j} = \begin{cases} p_{i,2j}^{SE} = \sin(i/10000^{2j/d}) \\ p_{i,2j+1}^{SE} = \cos(i/10000^{2j/d}). \end{cases} \tag{10}$$

with $j \in \{1, ..., d\}$, which represents the number of channels.

### C.2 EVALUATION METRICS AND SAMPLING QUALITY

To evaluate the generative performance of DeFoG, we adopt the V,U,N metrics, in accordance with the literature (Qin et al., 2025),

- *Validity* (V): The fraction of generated graphs that are accurately distributed. What is considered the valid structural property depends on the dataset.
- *Novelty* (N): The fraction of valid graphs not present in the training set. This metrics accounts for a measure of the memorization effect in the generated set.
- *Uniqueness* (U): The fraction of graphs that can not be reach by permuting another graph of the generated set.

We define the overall performance as the product VUN. This composite metric ensures that the model provides accurate, diverse, and non-redundant graphs. To achieve optimal generative performance, a model must then navigate between two failure modes: the *memorization phase*, where novelty of the generated samples collapses, and the *asymmetric phase*, where the uniqueness of the generated samples collapses due to equivariance breaking. Our goal is therefore to accelerate accuracy gains in early training while preventing a subsequent drop in U or N that would signify a transition into a memorization or asymmetric regime.

Lastly, we use the *Ratio* metrics (Table 1). This metrics is computed by aggregating structural graphs properties, such as node degrees, clustering coefficients, orbit count, eigenvalues of the normalized graph Laplacian and wavelet graph transform (Qin et al., 2025). The distance between each of these statistics and the empirical distribution of the generated graphs is then calculated using Maximum Mean Discrepancy, and finally aggregated into the *Ratio* metric. A *Ratio* of 1 is the shortest distance between train and test sets, and represents optimal generation.

## C.3 DATASETS

The analysis focuses on the *Stochastic Block Model* dataset (Martinkus et al., 2022), containing synthetic community graphs, where intra-community connection probabilities are higher than inter-community ones, $p_{intra} > p_{extra}$.

Two other random graphs models are investigated in Appendix G

- The *Erdös-Rényi* model, which generates graphs with node number between $n = 20$ and $n = 80$. Given $n$, the edges are independently sampled with constant probabilities $p = 0.6$. The adjacency matrix is hence sampled from

$$A_{i,j} \sim \text{Bernoulli}(p), \quad \forall i \neq j, \tag{11}$$

  where $A_{i,j}$ is also enforced to be symmetric, as we work with undirected graphs.

- The *Barábasi-Albert* model is a preferential attachment model, for example used to simulate citation networks. We use a fixed number of nodes $n = 64$, and graphs are sampled iteratively using the following algorithm. For each node $i$, $m = 6$ links will be established with other existing nodes. The probability of an edge to an existing node $j$ is given by its degree, $A_{i,j} \sim \deg(j)$. We then also ensure that the adjacency matrix is symmetric, for undirected graph generation.

In accordance with the literature of DeFoG (Qin et al., 2025), as well as, (Carballo-Castro et al., 2025), we also explore other datasets with distinct topological properties. Among them, the *planar* dataset (Martinkus et al., 2022), which consists of graphs that can be drawn on a plane without edge crossings. Planar graphs hence share a topological property, as opposed to *SBM*, graphs that instead stem from the same statistical distribution. We also investigate *Tree* dataset, that contains connected graphs with no cycles (Bergmeister et al., 2024).

The efficiency of our method on *Planar*, *Tree*, *Erdös-Rényi* and *Barábasi-Albert* is investigated in Appendix G.

## D TIME-DEPENDENT PERMUTATION RATES

We provide details on time-dependent permutation rates, as introduced in Section 2, and motivate the choices of rate functions $\chi(t)$ of Figure 3.

The rate function used in Figure 3 are shown on Figure 4(A). They show the time-evolution of the number of training steps between each permutations of the input graphs.

The choice of these rate functions is based on two observations. First, we noticed that above a certain threshold rate, the dynamics was greatly perturbed by the permutations, and the performance compromised. This effect can be seen in Figure 4(B), where the equivariant phase and the broken phase can be seen. On this diagram, green dots represent configuration with UN = 1 during the first 10000 training steps, and the red dots with UN < 1. Configurations are assumed stable if they have smaller values of $\chi$, or larger values of $\lambda$ than a measured stable state. The converse holds for red

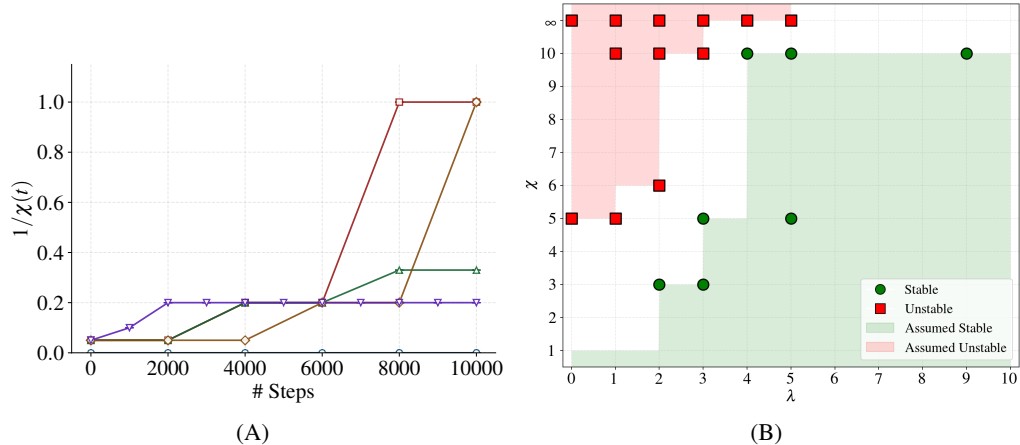

(A)                        (B)

Figure 4: (A) Inverse of the time-dependent rate functions used in Figure 3. (B) Phase space diagram of the aggregated metric UN, evaluated in the first 10000 training steps.

points. We noticed that when the rate functions $\chi(t)$ traverse the separation of phases, the validity of the graphs drops. Second, we observed that dynamics with smoother rate functions led to higher VUN performances.

## E  DATA AUGMENTATION

We investigate the interplay between data augmentation and sinusoidal positional encodings (Appendix C), on the *SBM* dataset. To this end, we augment the training graphs by sampling random permutations, such that the size of the dataset is increased by a factor of 3. In Figure 5, we compare the effect of positional encodings with and without data augmentation. We observe no benefit to data augmentation, when combined with symmetry breaking positional encodings. However, the interference between positional encodings and data augmentation is not clear yet, and is left for future work.

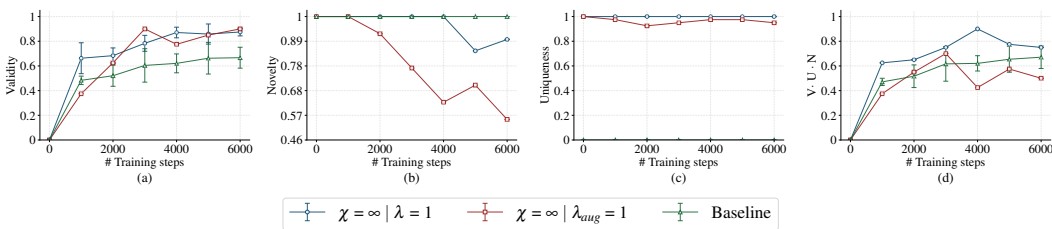

Figure 5: Comparison of positional encodings with and without data augmentation (a) Validity (b) Novelty (c) Uniqueness (d) VUN, on the *SBM* dataset.

## F  NORMALIZED SINUSOIDAL POSITIONAL ENCODINGS

In Section 2.2, we studied the dynamics of DeFoG using the positional encodings defined in Equation 10, for different values of $\lambda$. This definition of encodings results in larger values of $\lambda$ hence increasing the average value of the nodes label. However, this can lead to a delay of the learning of the model, especially when graphs are also permuted (Figure 2(C)). To tackle this issue, we suggest to normalize the positional encodings,

$$p_i(\lambda) = \frac{1}{\lambda}\left(\lambda\langle p\rangle_i + (p_i - \langle p\rangle_i)\right), \tag{12}$$

such that the average is independent of $\lambda$, $\langle p_i(\lambda) \rangle = \langle p \rangle_i$.

In Figure 6, we report different dynamics of DeFoG, with the positional encodings defined in Equation 12, for different values of $\lambda$, that we denote as $\lambda_s$ to differentiate it from Equation 10. Surprisingly, we notice that using normalized sinusoidal positional encodings (Equation 12), the collapse of Uniqueness and Novelty is further pushed back, compared the the non-normalized encodings (Equation 10), shown on Figure 2(B).

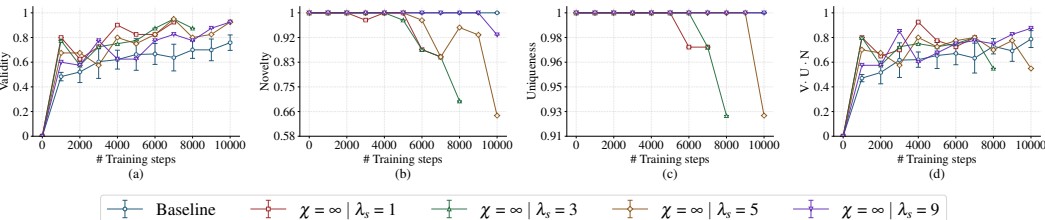

Figure 6: Comparison of DeFoG's baseline setting, with RRWP encodings and normalized symmetry breaking sinusoidal PE (Equation 12 with different invariance scaling $\lambda$ and no permutations, $\chi = \infty$ for (a) Validity (b) Novelty (c) Uniqueness (d) VUN, on the *SBM* dataset.

To further compare this normalization with the encodings introduced in Equation 10, we fix a value of $\lambda$, and set side by side the different methods presented in Section 2.1, as shown in Figure 7. In particular, the normalized encodings, denoted by $\lambda_s$, are compared to different permutations rates, $\chi$. This shows that the convergence of normalized encodings can compete with the encodings of Equation 10, but leads to an earlier collapse of UN. As matter of fact, the average of the encodings first introduced in Equation 10, scales with $\lambda$. This contrasts with the normalization of Equation 12 that preserves the average $\langle p_i \rangle$. This difference can prevent the equivariance breaking signal to be swamped by the constant node labels, potentially explaining the delayed UN collapses.

Lastly, we show the result of only providing a fraction of the nodes with positional encodings, here 75%, as denoted by $\lambda_{0.75}$ (Figure 7). This results show that this method does not allow the model to learn as efficiently, and motivates the focused on the analysis of the $(\lambda, \chi)$-plane.

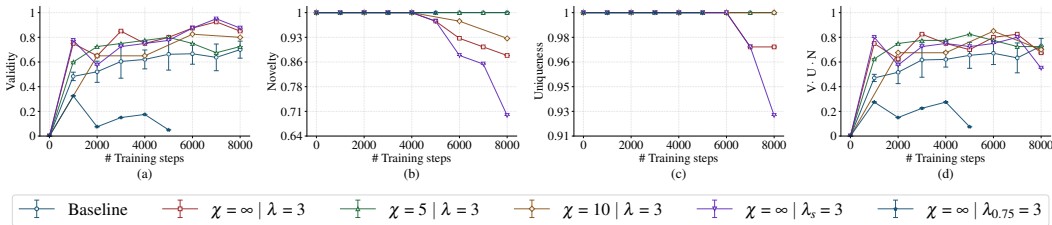

Figure 7: Comparison of the different methods introduced in Section 2.1 for a fixed value of $\lambda = 3$.

## G    TESTING THE METHOD ON OTHER DATASETS

We complement this analysis by showing how the method applies to other datasets. In particular, we compare the efficiency of sinusoidal positional encodings (Eq. 10) to the baseline, which instead uses RRWP encodings. The datasets are split in two categories. First, we consider *planar* and *tree* graphs, possessing a common topological property. Second, we assess the efficiency of the method on two random graphs models, *Erdös-Rényi* and *Barábasi-Albert* models. The different datasets are defined in Appendix C.3 and their generation parameters summarized in Table 2

### G.1    PLANAR GRAPHS

In Figure 8, we observe that the "learning window", as defined in Section 2, is not present for this dataset. As a matter of fact, uniqueness and novelty start to decrease very close to the peak

Table 2: Training and sampling parameters for the datasets used in Appendix G.

| Dataset | Min Nodes | Max Nodes | Graphs Sampled |
|---------|-----------|-----------|----------------|
| Planar | 64 | 64 | 40 |
| Tree | 64 | 64 | 40 |
| Erdös-Rényi | 20 | 80 | 40 |
| Barábasi-Albert | 64 | 64 | 40 |

of Validity, which results in a low (or even zero), VUN metric. Our analysis suggests that the coincidence of these V increase and U, N decrease is hard to disentangle, as at least different values of $\lambda$ always resulted in the same conclusion.

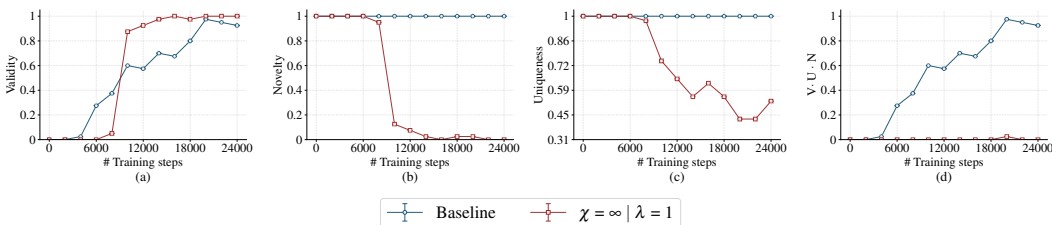

Figure 8: Comparison of DeFoG's baseline setting, with RRWP encodings and symmetry breaking sinusoidal PE (a) Validity (b) Novelty (c) Uniqueness (d) VUN, on the *planar* dataset.

## G.2 TREE GRAPHS

The method is further applied to Tree graphs, and the result is displayed on Figure 9. The conclusion is the same as for planar graphs (Appendix G.1), as the model does not initiate learning before U and N collapses. We conclude that the method, as introduced in this study, does not apply well on Tree graphs.

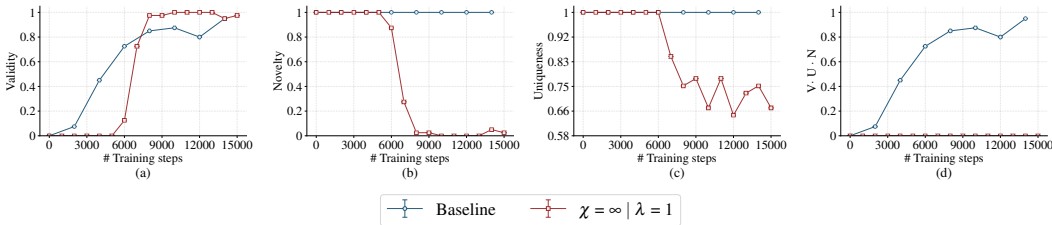

Figure 9: Comparison of DeFoG's baseline setting, with RRWP encodings and symmetry breaking sinusoidal PE (a) Validity (b) Novelty (c) Uniqueness (d) VUN, on the *tree* dataset.

## G.3 *Erdös-Rényi* MODEL

When applied to *Erdös-Rényi* graphs, however, the performance of sinusoidal encodings appears to match the baseline, as shown on Figure 10. This result also shows that U and N do not collapse, even in the presence of sinusoidal encodings, which for the *SBM* dataset where responsible for breaking these metrics (see Section 2.2). Since the performance of the baseline on this dataset is close to saturating the metrics, a potential performance gain from sinusoidal encodings is hard to assess. However, this method could provide a computational cost reduction, as the RRWP encodings are heavier to compute, and dependent on the graph structure.

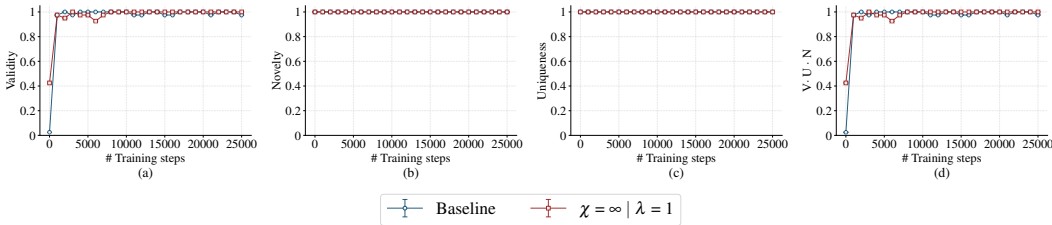

Figure 10: Comparison of DeFoG's baseline setting, with RRWP encodings and symmetry breaking sinusoidal PE (a) Validity (b) Novelty (c) Uniqueness (d) VUN, on the *Erdös-Rényi* model.

### G.4  *Barábasi-Albert* MODEL

Similarly, when used on *Barábasi-Albert* graphs, our method shows a similar performance with the baseline, as can be seen on Figure 11. However, similarly to the *Erdös-Rényi* model (Appendix G.3), the baseline saturates the VUN metrics. We hence do not provide a comparative study of the performance of our method, with the baseline, but merely state the computational benefit stemming from the simple, light to compute and graph-independent sinusoidal positional encodings.

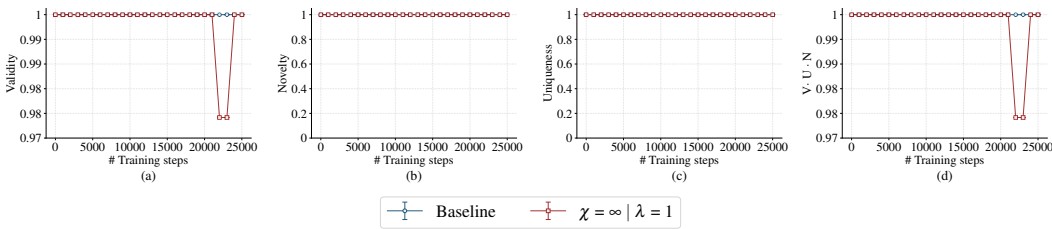

Figure 11: Comparison of DeFoG's baseline setting, with RRWP encodings and symmetry breaking sinusoidal PE (a) Validity (b) Novelty (c) Uniqueness (d) VUN, on the *Barábasi-Albert* model.

**Conclusion**    These results delimit the scope of our method's effectiveness. While we do not offer a definitive explanation for the performance drop of *planar* and *tree* dataset, we note that these graphs are uniquely defined by their topology. We hypothesize that capturing such structural properties requires symmetry-aware positional encodings, such as RRWP. Conversely, on random graph models like the SBM, sinusoidal encodings remain competitive with RRWP.

